# Evaluation of surveillance-response interventions for *Schistosoma haematobium* elimination on Pemba Island, Tanzania: A 4-year intervention study with repeated cross-sectional surveys

Lydia Trippler[1,2,3], Jan Hattendorf[1,2], Mohammed Nassor Ali[4], Sarah Omar Najim[4], Khamis Seif Khamis[5], Khamis Rashid Suleiman[4], Shaali Makame Ame[6], Saleh Juma[5], Fatma Kabole[6], Said Mohammed Ali[4], Stefanie Knopp[1,2]*

1 Swiss Tropical and Public Health Institute, Allschwil, Switzerland, 2 University of Basel, Basel, Switzerland, 3 University of Glasgow, Glasgow, Scotland, United Kingdom, 4 Public Health Laboratory – Ivo de Carneri, Chake Chake, Pemba, United Republic of Tanzania, 5 Neglected Diseases Program, Zanzibar Ministry of Health, Chake Chake, Pemba, United Republic of Tanzania, 6 Neglected Diseases Program, Zanzibar Ministry of Health, Zanzibar Town, Unguja, United Republic of Tanzania

* s.knopp@swisstph.ch

## Abstract

### Background

The Zanzibar islands, Tanzania, have eliminated schistosomiasis as a public health problem since 2017 and need to rethink their intervention strategies to ensure that the progress made is sustained and advanced. We evaluated the performance of a novel surveillance-response approach for interrupting *Schistosoma* transmission on Pemba Island from 2020-2024.

### Methodology

In low-prevalence implementation units (IUs), surveillance-response interventions were implemented, which consisted of active and reactive case finding, treatment of *S. haematobium*-positive individuals, and reactive snail control. The performance of the surveillance-response interventions was measured by sensitivity, timeliness and impact on prevalence. Annual cross-sectional surveys were conducted in schools and households to estimate the total number of individuals infected with *S. haematobium* in the area and the proportion identified by the surveillance-response approach. Uro-genital schistosomiasis was diagnosed by egg microscopy.

### Principal Findings

Among the 20 IUs in the study area, 15, 16, and 17 were considered low-prevalence IUs in the intervention periods in 2021, 2022, and 2023, respectively. Across the intervention periods, 4.6% (707/15509) among the schoolchildren included in active surveillance were tested *S. haematobium*-positive and treated. During

**Data availability statement:** All relevant data are within the manuscript and its Supporting Information files.

**Funding:** Funding for the study has been obtained from the Swiss National Science Foundation (SNSF; Bern, Switzerland) via a PRIMA grant (PR00P3_179753) of SK. The funders had no role in study design, data collection and analysis, decision to publish, or preparation of the manuscript.

**Competing interests:** The authors have declared that no competing interests exist.

reactive surveillance, at water bodies 8.2% (10/122) and in households 9.9% (45/454) of individuals were found infected and treated. Moreover, 47 among the 262 waterbodies were treated with molluscicide. The overall sensitivity of the surveillance-response approach across 2 periods, where complete surveillance data were available, was 23.0%. The timeliness of reactive interventions was 2 weeks. In the low-prevalence IUs, the prevalence in schoolchildren changed from 0.5% (7/1552) in 2021 to 0.4% (6/1653) in 2022, from 0.6% (12/2123) in 2022 to 0.7% (15/2240) in 2023, and from 0.4% (8/2287) in 2023 to 1.0% (27/2755) in 2024 after surveillance-response implementation. The respective prevalence in community members was 0.5% (14/2969) in 2021 and 0.7% (19/2928) in 2022, 0.6% (18/3175) in 2022 and 0.3% (10/2979) in 2023, and 0.4% (12/3257) in 2023 and 0.7% (22/3106) in 2024.

## Conclusion

Surveillance-response interventions maintained the low *S. haematobium* prevalence, but interruption of transmission was not achieved. The overall sensitivity of the approach was low. Timeliness was very good but required strong communication and collaboration between the surveillance-response teams. To work on a larger scale, with good coverage and improved sensitivity, elimination programs will need a large number of well-trained staff and adequate tools for surveillance and response activities, data management, and communication.

## Trial registration

ISRCTN, ISCRCTN91431493. Registered 11 February 2020, ISRCTN - ISRCTN91431493: Novel intervention strategies for schistosomiasis elimination in Zanzibar.

## Author summary

Schistosomiasis is a neglected tropical disease that is caused by infection with *Schistosoma* parasites, which are transmitted in natural freshwater bodies. On the Zanzibar islands, Tanzania, intense interventions, including large-scale treatment campaigns, have successfully reduced the number of *Schistosoma* infections and associated morbidity over the past decades. To fully eliminate schistosomiasis from Zanzibar, new intervention strategies need to be explored. Here, we assessed the performance of a novel targeted intervention approach termed "surveillance-response" in low-endemic areas on Pemba Island. We screened children for schistosomiasis in schools, treated the positives and tracked them to their homes and to freshwater bodies that they used, where we also tested present individuals and treated the positives. At the water bodies, we conducted surveys to detect intermediate host snails and sprayed a molluscicide if snails were present. In annual surveys, we found that the surveillance-response

interventions maintained the already low number of schistosomiasis cases, but complete elimination was not achieved within 4 years. The ability of the approach to detect all schistosomiasis cases was low, but the timeliness from start to end of the procedures was very good. To conduct surveillance-response at a larger scale with good population coverage and improved sensitivity for case detection, schistosomiasis elimination programs will need a large number of well-trained staff and adequate tools for surveillance and response activities, data management, and communication.

## Introduction

Schistosomiasis is a neglected tropical disease that affects more than 250 million people worldwide [1,2]. The global burden due to schistosomiasis was estimated at 1.86 million disability adjusted life years (DALYs) in 2021 [3]. Most people affected by schistosomiasis live in Africa, where the burden was estimated at 1.57 million DALYs in 2021 [3].

The World Health Organization (WHO) has set the goal to eliminate schistosomiasis as a public health problem globally and eliminate infections in humans in specific countries by 2030 [4,5]. Mass drug administration (MDA) with praziquantel has long been considered the primary strategy for controlling and eliminating schistosomiasis [6]. Despite its success in reducing prevalence in many countries, MDA has limitations, including concerns about potential drug resistance, increasing treatment hesitancy in target populations due to the lack of prior testing and fear of side effects, and evolving challenges related to restricted drug availability and limited funding for the implementation of MDAs [5,7–10]. Moreover, in areas that have successfully controlled schistosomiasis and achieved elimination as a public health problem (<1% of heavy intensity infections), only a few people are infected with *Schistosoma* parasites, and hence, only a small part of the population requires treatment [11–13].

In their new monitoring and evaluation framework for schistosomiasis, published in 2024, the WHO flags that areas that have achieved elimination as a public health problem will enter a maintenance phase where a low prevalence of infection may still persist and where MDA at a reduced frequency may be necessary to maintain program gains and prevent recrudescence of infection [4,14]. The WHO also indicates that surveillance will be essential until there is no more risk of a rebound of prevalence and intensity of infection [4,14].

While details of a surveillance approach for schistosomiasis were not yet provided by the WHO in late 2025, general guidelines for communicable disease surveillance-response systems were published two decades ago [14]. Surveillance-response is used as an intervention in other disease elimination programs, for example malaria [15]. Also for schistosomiasis, it has long been pointed out that to ensure that the progress made by control programs is sustained and advanced, countries and areas in the elimination phase must strengthen their health systems and participate in the development and use of integrated surveillance-response schemes [16]. It is suggested that surveillance, followed by public health actions consisting of response packages tailored to interruption of transmission in different settings, will help to effectively achieve the schistosomiasis control and elimination goals set by WHO for 2030 [5,16,17].

The Zanzibar Islands of the United Republic of Tanzania look back to a long history of research and interventions against urogenital schistosomiasis [18]. Over the past decades, the prevalence of and morbidity due to *S. haematobium* were substantially reduced [18]. Elimination efforts started in 2012, with the Zanzibar Elimination of Schistosomiasis Transmission (ZEST) project [13,19], where, in addition to island-wide MDA in schools and communities, also behavior change communication and snail control interventions were implemented [13,20]. In 2017, in many areas of the islands, urogenital schistosomiasis had been eliminated as a public health problem [18,20,21]. The endemic landscape is nowadays very heterogeneous: most communities on the island have a very low or zero prevalence, but in some areas higher prevalences persist or rebound quickly after interventions [21]. Large-scale MDA across the islands seems no longer justified, and new targeted intervention approaches are needed to ensure that the progress made is sustained and/or interruption of transmission accelerated. The SchistoBreak study, implemented on Pemba Island from 2020 to 2024,

tested a novel adaptive intervention approach that consisted of i) a comprehensive intervention package including MDA, snail control and behavior change communication in hotspot areas [22–24] and ii) novel surveillance-response interventions including active, passive and reactive case finding, treatment of *S. haematobium*-positive individuals, and reactive snail control in low-prevalence areas. Here, we evaluated the performance of the surveillance-response approach for its sensitivity, timeliness and impact on prevalence.

## Methods

### Ethics

The SchistoBreak study protocol was waived by the ethics committee in Switzerland (Ethikkommission Nordwest- und Zentralschweiz; EKNZ) on October 23, 2019 (Req-2019–00951) and received annual ethical approval by the Zanzibar Health Research Institute (ZAHRI). The first approval was given on December 13, 2019 (ZAHREC/01/PR/December/2019/12), and the latest approval was given on March 31, 2023 (ZAHREC/04/AMEND/MARCH/2023/03). The study was prospectively registered at ISRCTN (ISCRTN91431493).

Twice a year, at the beginning of each study phase (annual cross-sectional surveys and intervention periods, respectively), all leaders (shehas) of the implementation units (IUs) and teachers of the schools involved in the study were invited to participate in information meetings at the Public Health Laboratory – Ivo de Carneri (PHL-IdC). Here, the aims and procedures of the study were explained, results were presented, and challenges for study implementation were discussed. Moreover, the shehas and teachers were asked to notify their community members and parents of the children about forthcoming activities of the study and to encourage their participation.

All participants of annual cross-sectional surveys and surveillance interventions were provided with information sheets about the study procedure, including the telephone number of the local principal investigator in case of upcoming questions. All participants were asked to provide written consent by signing an informed consent form. In the case of participating children <18 years, the legal guardians were asked to sign the informed consent form. In addition, children aged 12–17 years were invited to provide written assent for participation by signing an assent form.

### Study site

The SchistoBreak study was conducted on Pemba Island, which is part of the semi-autonomous Zanzibar archipelago of the United Republic of Tanzania. Pemba is situated approximately 30 km off the coast of mainland Tanzania and is divided into four districts, which are subdivided into 129 small administrative areas, called shehias [25]. In the SchistoBreak study, each shehia represented an IU. The study included 20 IUs in two districts in the north of Pemba, Micheweni and Wete. According to the 2022 Tanzanian census, the study area had a population of about 95,000 inhabitants and a population growth rate of 2.5 [25]. Of the 20 IUs, 18 IUs had at least one public primary school. Most of these primary schools also included a nursery school. Additionally, in each IU, there were several Islamic schools, called madrassas. Finally, a total of 22 health facilities were located in the study area, including 20 primary health care units (PHCUs) and two district hospitals [26].

The only *Schistosoma* species transmitted locally on the Zanzibar islands is *S. haematobium* [18]. Elimination of urogenital schistosomiasis as a public health problem, defined as <1% heavy-intensity *Schistosoma* infections, was reached in 2017 [18,20,21]. While the *S. haematobium* prevalence is very low or zero in most areas, some rebounding or persistent hotspot areas also exist [21].

### Study design and participants

The SchistoBreak study was designed as an intervention study with repeated cross-sectional surveys conducted in schools and households to monitor the prevalence of *S. haematobium* in the IUs and to assess the impact of interventions

[22]. The fieldwork for the study was implemented from February 2020 until March 2024. Annual cross-sectional parasitological surveys were conducted in schools and households of all 20 IUs of the SchistoBreak study from November to February/March each year. Based on the results of the cross-sectional surveys, IUs were classified as low-prevalence IUs when they had a *S. haematobium* prevalence of <3% in schoolchildren and <2% in community members in the surveys, and as "hotspot" IUs when they had a *S. haematobium* prevalence of ≥3% in schoolchildren and ≥2% in community members. Interventions were implemented between the annual surveys in all IUs from May to October each year. The interventions were targeted at either low-prevalence or hotspot IUs, as revealed in the annual surveys. In low-prevalence IUs, surveillance-response interventions were implemented, as described below. The multi-disciplinary interventions implemented in hotspot IUs are not the focus of this manuscript and are described elsewhere in detail [22,27]. To be eligible for participation in the cross-sectional surveys or intervention activities of this study, individuals had to meet the following criteria: i) living or attending school in one of the IUs, ii) being aged ≥4 years, and iii) submitting the required informed consent forms.

**Active and reactive surveillance-response interventions**

In all low-prevalence IUs of the study area, identified through the annual cross-sectional surveys, surveillance-response interventions were implemented in the intervention period following the survey for approximately 6–7 months. Surveillance-response interventions consisted of active, passive, and reactive surveillance and reactive snail control.

To start with active surveillance, in each low-prevalence IU, one primary school and one madrassa were selected. In each IU, the largest public primary school was chosen, defined as the school with the highest number of enrolled children. The madrassa was selected based on risk and ensuring a wide coverage of the study area, considering the following indicators: The madrassa had i) a minimum of 50 students, ii) was located max. 500 m distant from a water body, and iii) was the madrassa farthest away from the selected public primary school. Once the schools were visited, the procedure was explained to all children in grades 3–5 in the primary school and to all children in the madrassa. On the first day of the study team's visit, the children were registered and handed out information sheets and consent forms to be given to their parents. On the second day, once signed informed consent forms had been submitted, children's demographic data were recorded, and they were invited to provide their fresh urine sample.

All collected urine samples were tested for microhematuria using Hemastix reagent strips (Siemens Healthcare Diagnostics AG; Zürich, Switzerland) at the point of care in the schools or at the laboratory of the PHL-IdC. In addition, in 2021, all urine samples were tested for *S. haematobium* infection with a portable PCR machine (diaxxo AG; Zürich, Switzerland) at the point of care in the schools. Moreover, in 2021, urine samples that were tested positive for microhematuria or by the PCR machine, and a random selection of 20 (<500 children in school grades 3–5 or the madrassa) or 30 (≥500 children in school grades 3–5 or the madrassa) samples with a negative test result were examined for *S. haematobium* infection at PHL-IdC, using the urine filtration method. In the intervention periods of 2022 and 2023, in addition to microhematuria assessment conducted at the point of care, all collected urine samples were additionally examined for *S. haematobium* infection at PHL-IdC, using the urine filtration method. As a response, if a child tested positive for *S. haematobium* and/or microhematuria, they were treated with a single dose of praziquantel (40 mg/kg body weight), using a dose pole [28].

Subsequently, for reactive surveillance, the study team accompanied children who had a positive test result to their homes and to the freshwater bodies they had used in the past. At the child's household and the water bodies, all present individuals were invited to participate in the study and hence to answer a short questionnaire about demographics and to provide their fresh urine sample. The urine sample was tested for microhematuria using Hemastix reagent strips at the point of care in the household or at the water body, respectively, and for *S. haematobium* using urine filtration at PHL-IdC. Microhematuria-positive individuals were treated at the point of care with a single dose of praziquantel (40 mg/kg body weight). A positive *S. haematobium* test result was communicated to the individual by phone. The infected person was then advised to seek praziquantel treatment at the nearest health facility. Alternatively, a meeting was arranged with a study team member where praziquantel was provided.

### Reactive snail control intervention

For reactive snail control, an experienced study team visited all freshwater bodies indicated to be used by positive-tested children. Surveys were conducted at the water bodies to identify the presence of snails of the genus *Bulinus*, which is the intermediate host snail for *S. haematobium* on Pemba [18]. The team conducted surveys at the water bodies to collect data on the presence/absence of different snail species, water chemistry, the presence and type of water vegetation and human activities. At human water contact sites, where *Bulinus* were present, the molluscicide niclosamide (WP83,1; Bayer AG Crop Science Division) was applied to reduce the number of intermediate host snails. Depending on the size and nature of the water body, niclosamide was applied using either plastic backpack sprayers (Farmate, Taizhou Sunny Agricultural Machinery Co., LTd, Taizhou, China) or a petrol sprayer (Zhejiang O O Power Machinery Co., Ltd, Zhejiang, China). If no *Bulinus* were found at the present visit, but there was a history of *Bulinus* identified at earlier visits during the study, niclosamide was also applied. If there was no indication of *Bulinus* at a past or present survey, niclosamide was not applied. Details of snail surveys and control are provided elsewhere [23].

### Passive surveillance intervention

In addition to active and reactive surveillance in schools, madrassas, and communities, passive surveillance was implemented in local health facilities from 2020 to 2024. For passive surveillance, employees of all health facilities in the study area were invited twice a year to PHL-IdC for a meeting. During the meeting, an update on the SchistoBreak study was given, symptoms of schistosomiasis were explained, and the challenges of testing and treating schistosomiasis in health facilities and their solutions were discussed. Additionally, all health facility employees participating in the meeting received training on the use of Hemastix reagent strips and were explained how to fill out report forms for the SchistoBreak study. Finally, all health facility employees were provided with Hemastix reagent strips and praziquantel to test individuals presenting at the health facility with symptoms of urogenital schistosomiasis and to treat those with identified microhematuria. The health facilities were also equipped with reporting forms to record the demographic and symptom data of the patients tested with Hemastix reagent strips. Once a week, a member of the SchistoBreak study team visited the health facilities to collect the report forms and to assess whether the facilities still had an adequate supply of Hemastix reagent strips and praziquantel tablets.

### Cross-sectional parasitological surveys

To monitor the prevalence of *S. haematobium* and microhematuria across the study years and to classify the study area into low-prevalence or hotspot IUs for interventions, four annual cross-sectional parasitological school and community surveys were conducted from 2020 to 2024. The implementation procedures of the cross-sectional surveys in schools and communities are described elsewhere in detail [22,27,29].

In brief, for the school surveys, the largest public primary school in each IU was selected. On the first day of work in each school, 25 children per grade in seven grades (nursery, grades 1–6) were randomly selected and registered for the study [29]. The children were provided with an information sheet and informed consent form to be given to and signed by their parents. On the following day, once the selected children had submitted the signed informed consent form, the study team recorded their demographic details in a brief questionnaire interview. The children were then asked to collect their own urine sample in a transparent plastic container and to submit it to the study team.

For the community surveys, 80 (70 in 2021) housing structures in each IU were randomly selected using geographic vector data. Based on the geolocation of housing structures and using two combined mobile applications installed on tablet computers [29], field enumerators were able to locate the housing structures in the communities. If housing structures were inhabited, eligible household members were invited to participate in the survey. Once written consent was obtained, individuals were asked to provide a urine sample. One of the adult household members was invited to participate in a

questionnaire interview and provide demographic data for all eligible household members. If household members were not at home at the time of the visit, the plastic containers for urine collection were left with an instructed present individual and collected by the study team on the following day. All urine samples were stored in a cool box and transported back to the PHL-IdC for laboratory analyses.

## Laboratory procedures

During the cross-sectional surveys and surveillance interventions, urine samples were collected and transferred to the PHL-IdC on the day of collection. At the PHL-IdC, all urine samples were examined for microhematuria using Hemastix reagent strips, unless they had already been tested at the point of care during the intervention period. The intensity of microhematuria was graded according to the manufacturer's color chart into negative, trace, small, moderate, or large. Additionally, urine samples were subjected to the urine filtration microscopy method. For this purpose, 10 ml of urine were passed through a 13 mm fabric filter (Sefar Ltd., Bury, United Kingdom), which was placed in a Swinnex plastic filter holder (Millipore, Merck KGaA, Darmstadt, Germany), using a plastic syringe. Subsequently, the filters were examined for the presence and quantity of *S. haematobium* eggs using a light microscope. Participants with 1–49 eggs per 10 ml of urine were considered to have a light-intensity infection, while individuals with 50 or more eggs per 10 ml of urine were considered to harbor a heavy-intensity infection [30].

## Data management

Demographic data collected during registration and questionnaire interviews in the cross-sectional surveys and intervention periods in schools, households, and at water bodies were recorded using Open Data Kit (ODK, www.opendatakit.org) installed on Samsung Galaxy Tab A tablets. The electronic data were sent to the secured ODK server hosted by the Swiss Tropical and Public Health Institute (Swiss TPH) in Allschwil, Switzerland. Results of Hemastix reagent strip diagnostics conducted at the point of care in schools, households, and at water bodies during the intervention period were also captured with ODK. Additionally, urine filtration results from the intervention period of 2021 were recorded using ODK. Urine filtration results of all four cross-sectional surveys and the intervention periods in 2022 and 2023, respectively, were captured on paper and subsequently double-entered in Microsoft Excel spreadsheets (version 2016) by two experienced data entry clerks. Any double-entry mismatches were verified against the original paper forms and corrected. For statistical analysis, the laboratory results were merged with the questionnaire and registration data. To inform participants about their *S. haematobium* infection status, participant names were matched with laboratory analysis data. Otherwise, names were kept in a separate file, and only coded data were used for statistical analyses. All data cleaning was conducted using R and STATA/IC 16.1.

## Statistical analysis

To demonstrate the impact of the surveillance-response interventions, the prevalence of *S. haematobium* infections and microhematuria in the low-prevalence IUs was calculated per year, based on the annual cross-sectional school and community surveys.

To evaluate the surveillance-response approach, its sensitivity was calculated per age group (children or adults), per year, and overall, using urine filtration microscopy results as the diagnostic reference standard. The sensitivity of surveillance interventions for infectious diseases is defined as "the proportion of cases of a disease or health condition detected by the surveillance system" [14,31]. The sensitivity of the surveillance-response approach was determined with the following formula:

$$Sensitivity = \frac{N_{inf\ detected\ by\ surveillance}}{N_{inf}}$$

The total number of individuals infected with *S. haematobium* in the study area ($N_{inf}$) was unknown and was estimated with the following formula:

$$N_{inf,i} = \left( \frac{P_{i,\ children} + P_{i+1,children}}{2} \times N_{children} \right) + \left( \frac{P_{i,\ adults} + P_{i+1,adults}}{2} \times N_{adults} \right)$$

Whereby the *S. haematobium* prevalence (*P*) is based on the urine filtration microscopy results of repeated cross-sectional parasitological school and community surveys, and *i* represents the year of intervention. As the surveillance-response interventions were conducted midway between two annual cross-sectional surveys, the average prevalence of the survey before and the survey after the intervention phase was used as the reference prevalence, respectively ($N_{inf,2021}$ = estimated infections in 2021, $N_{inf,2022}$ = estimated infections in 2022, and $N_{inf,2023}$ = estimated infections in 2023). The population size (*N*) of the study area was determined based on data obtained from the 2022 Tanzania population census [25]. Based on the population growth rate of 2.5, the population sizes for the years 2021, 2023, and 2024 were estimated. To determine the number of school-aged children and adults in the population, percentages from the 2012 Tanzania population census were used [32], since the socio-demographic detailed numbers from the 2022 census were not yet publicly available. It was estimated that 40.7% of the population in the study area was aged 4–17 years, 45.3% were aged ≥18 years, and 14.0% were aged <4 years. The latter population group was not considered in the analysis for the sensitivity, since this age group was not included in the study.

To assess robustness of the sensitivity of the surveillance approach, we varied two key assumptions in a sensitivity analysis: i) estimated population sizes were scaled by factors of 0.8, 0.9, 1, 1.1 (representing -20%, -10%, 0%, 10% change relative to projected census estimates), and ii) mid-period prevalence was scaled by factors of 1, 0.75, 0.5 (representing reductions of 0%, 25%, 50% relative to the cross-sectional mean prevalence), reflecting uncertainty in population denominators and lower true prevalence at the time of surveillance due to post-survey treatment.

For the calculation of $P_{school}$ in 2021, multiple imputation was performed for missing urine filtration data using the mi package in R, and prevalence estimates were pooled. The imputation model included microhematuria results, results from the portable PCR machine, age, and sex.

For the total count of individuals tested during reactive surveillance in households and at water bodies, only individuals who were identified based on positive *S. haematobium* or microhematuria results of index children were included in the analyses. Individuals in households and at water bodies who were identified solely based on a child's positive PCR test result in 2021 were not included in the analysis.

The timeliness of the surveillance-response approach was evaluated, defined as the reflection of "speed or delay between steps in a surveillance system" [31]. For the analysis, the time between the registration of children and the following steps were assessed: i) microhematuria testing, ii) urine filtration testing, iii) treatment, iv) registration of household members, v) registration of individuals at water bodies, and vi) reactive snail control. For assessment of timeliness, negative time values were observed between two steps, such as when a water body had already been followed up and/or surveyed for snails based on one child's information, and was then indicated by another child as well. In such cases, the negative time values were treated as zero.

## Results

### Identification and treatment of *S. haematobium* cases through active surveillance

Based on the annual cross-sectional school and community surveys conducted in the 20 IUs of the SchistoBreak study in 2021, 2022, and 2023, 15 IUs were considered low-prevalence IUs in the intervention periods in 2021, 16 in 2022, and 17 in 2023, respectively.

At baseline in 2021, 11 of the 15 low-prevalence IUs had a public primary school, where active surveillance was implemented (Fig 1). Furthermore, active surveillance was conducted in 15 madrassas. In 2022, 14 of the 16 low-prevalence

IUs had public primary schools, where active surveillance started, in addition to 16 madrassas. In 2023, 15 of the 17 low-prevalence IUs had a public primary school, where active surveillance was implemented. However, one of these 15 schools had only recently opened, and only its nursery school and grades 1 and 2 were running. Since active surveillance was restricted to grades 3–5, only 14 public primary schools were part of the interventions. In the same year, active surveillance was also conducted in 17 madrassas.

For active surveillance, in 2021, a total of 3703 children were tested for *S. haematobium* and/or microhematuria in primary schools, of whom 214 (5.8%) were found positive. In madrassas, 56/592 (9.5%) children had a positive test result. In 2022, a total of 128/4455 (2.9%) children from primary schools and 25/817 (3.1%) children from madrassas were tested for *S. haematobium* and/or microhematuria in primary schools and identified as positive. In 2023, a total of 245/4928 (5.0%) children from primary schools and 39/1014 (3.8%) from madrassas had a positive test result. Across the intervention periods, a total of 707 children tested positive for *S. haematobium* and/or microhematuria, who were subsequently tracked to their households and to the water bodies they frequented as part of reactive surveillance (Fig 1). Table 1 shows the socio-demographic characteristics of the participants in active and reactive surveillance in low-prevalence IUs in the three study periods.

### Identification and treatment of *S. haematobium* cases and water bodies with *Bulinus* through reactive surveillance

In 2021, tracking children who were infected with *S. haematobium* and/or had microhematuria in urine through active surveillance to water bodies they had used, resulted in an additional 67 individuals who were followed up at water bodies. Among them, urine filtration results were available for 66, of whom 5 (7.6%) tested positive for *S. haematobium*. In addition, 243 individuals were followed up in households. Of these, 210 individuals had urine filtration results, with 28 (13.3%) testing positive for *S. haematobium*. In 2022, 22 individuals were followed up at water bodies, with urine filtration results available for 21 of them. Among these, 2 (9.5%) tested positive for *S. haematobium*. Furthermore, 104 individuals were followed up in households. Of these, urine filtration results were available for 93 individuals, of whom 5 (5.4%) tested positive for *S. haematobium*. In 2023, based on the tracked index cases from schools, 47 individuals were followed up at water bodies. For 35 of these, urine filtration results were available, of which 3 (8.6%) tested positive. Furthermore, 152 individuals were followed up at households. Among these, urine filtration results were available for 151, of whom 12 (7.9%) individuals in households were infected with *S. haematobium*. Across all years, 136 individuals were followed up at water bodies with 122 urine filtration results available, of which 10 (8.2%) were tested positive (Fig 2A). Furthermore, 499 individuals were followed up in households with 454 urine filtration results, of whom 45 (9.9%) individuals were infected with *S. haematobium* (Fig 2B).

In the intervention period of 2021, 112 water bodies were visited for reactive malacological surveys, and snails of the genus *Bulinus* were detected in 10 (8.9%) of them (Fig 2A and Table 2). In a total of 13 (11.6%) water bodies, snail control with the molluscicide niclosamide was conducted. In 2022, 67 water bodies were visited for reactive malacological surveys, and among them, *Bulinus* were detected in 15 (22.4%). Snail control was conducted in a total of 19 (28.4%) water bodies. In 2023, 83 water bodies were visited for reactive malacological surveys, *Bulinus* were detected in 10 (12.0%) water bodies, and 15 (18.1%) were treated with niclosamide.

### Sensitivity of the surveillance-response approach in the low-prevalence implementation units

Based on the prevalence of the cross-sectional school and community surveys conducted in 2021 and 2022, and population size estimations from the national census, a total of 241 individuals were estimated to be infected with *S. haematobium* in the 15 low-prevalence IUs in 2021 (Fig 3). During the first surveillance-response intervention period in 2021, 150 school-aged children tested positive for *S. haematobium* based on urine filtration microscopy conducted in a subgroup of school-aged children in the 15 low-prevalence IUs. However, the data imputation indicated that a total of 270 school-aged

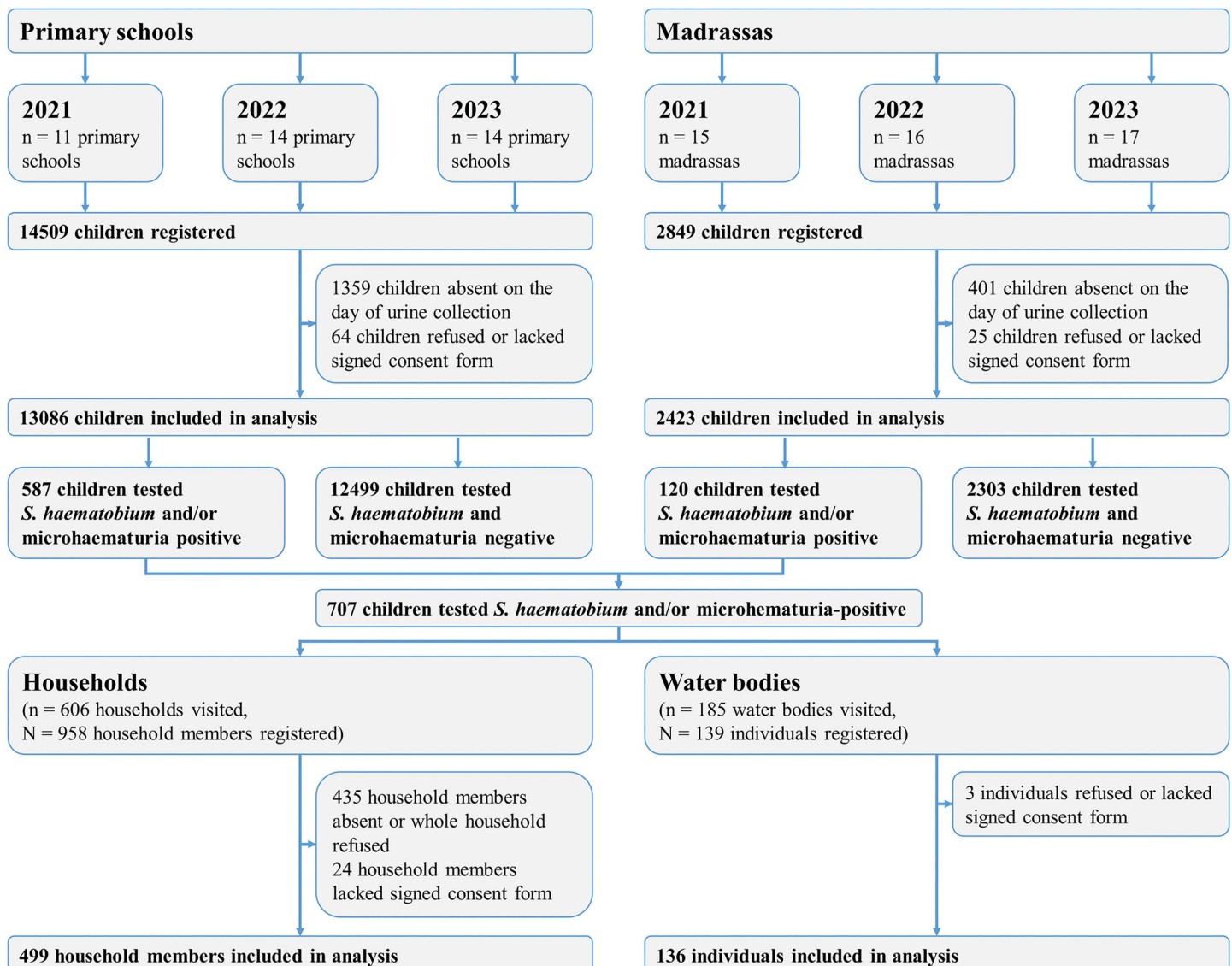

**Fig 1. Flow diagram of individuals participating in active and reactive surveillance of the SchistoBreak study from 2021 to 2023 in the north of Pemba, Tanzania.**

children would have tested positive if we had tested all samples for *S. haematobium* using urine filtration microscopy. Based on the imputed number of 270 positive school-aged children, the overall sensitivity of the surveillance interventions in 2021 was 114.2%, and varied between 103.8% and 285.5% across sensitivity analysis scenarios (Table 3). In 2022, a total of 327 individuals were estimated to be infected with *S. haematobium* in the 16 low-prevalence IUs, based on the prevalence data retrieved from the school and community surveys conducted in 2022 and 2023. During the second intervention period in 2022, 48 individuals tested *S. haematobium*-positive based on urine filtration in the 16 low-prevalence IUs. Hence, the overall sensitivity of the surveillance-response interventions was 14.7%, and varied between 13.4% and

**Table 1. Socio-demographic characteristics of the participants in active and reactive surveillance in low-prevalence implementation units on Pemba from 2021 to 2023.**

| | | 2021 | | | 2022 | | | 2023 | | |
|---|---|---|---|---|---|---|---|---|---|---|
| | Total (N=15907) | School (N=4295) | HH (N=167) | WB (N=67) | School (N=5272) | HH (N=40) | WB (N=22) | School (N=5942) | HH (N=55) | WB (N=47) |
| **Sex** | | | | | | | | | | |
| Female | 8436 (52.3%) | 2224 (51.8%) | 142 (58.4%) | 32 (47.8%) | 2786 (52.8%) | 57 (54.8%) | 6 (27.3%) | 3082 (51.9%) | 93 (61.2%) | 14 (29.8%) |
| Male | 7708 (47.7%) | 2071 (48.2%) | 101 (41.6%) | 35 (52.2%) | 2486 (47.2%) | 47 (45.2%) | 16 (72.7%) | 2860 (48.1%) | 59 (38.8%) | 33 (70.2%) |
| **Age (years)** | | | | | | | | | | |
| Median [Min, Max] | 11.0 [4.0, 100] | 11.0 [4.0, 29.0] | 16.0 [4.0, 100] | 14.0 [6.0, 72.0] | 11.0 [4.0, 23.0] | 18.0 [4.0, 65.0] | 13.0 [6.0, 62.0] | 11.0 [4.0, 19.0] | 25.0 [4.0, 65.0] | 11.0 [4.0, 69.0] |

School = Active surveillance in primary schools and madrassas; HH = Reactive surveillance in households of *S. haematobium* and/or microhematuria-positive children; WB = Reactive surveillance at water bodies used by *S. haematobium* and/or microhematuria-positive children.

36.8% across sensitivity analysis scenarios. In 2023, 443 individuals were estimated to be infected with *S. haematobium* in the 17 low-prevalence IUs. During the intervention period, 129 individuals tested *S. haematobium*-positive based on urine filtration, resulting in a sensitivity of the surveillance interventions of 29.1%, which varied between 26.4% and 72.8% across sensitivity analysis scenarios. Across the three intervention periods (2021, 2022, and 2023), including partially imputed data for school-aged children in 2021, the overall sensitivity of the surveillance interventions was 44.8%, with sensitivity estimates ranging from 40.7% to 112% across the combined sensitivity analyses scenarios. Across the two intervention periods (2022 and 2023), for which complete urine filtration data were available for children and adults, and no data were imputed, the overall sensitivity of the surveillance interventions was 23.0%, with sensitivity estimates ranging from 20.9% to 57.5% across the combined sensitivity analyses scenarios (Table 3). The sensitivity of the surveillance interventions to detect *S. haematobium*-infected school-aged children was 234.4% in 2021 (where data were partially imputed), 22.3% in 2022, and 53.5% in 2023. The sensitivity of the surveillance interventions to detect *S. haematobium*-infected adults was 4.2% in 2021, 2.0% in 2022, and 2.8% in 2023.

### Timeliness of surveillance-response interventions in low-prevalence implementation units

As shown in Fig 4, in the low-prevalence IUs, the median number of days between the registration of children in primary schools or madrassas and the examination of their urine samples for microhematuria and *S. haematobium* infection, respectively, was one day (range: 0–12 days and 0–18 days, respectively). The median number of days between the registration in school and the praziquantel treatment of positive children was seven days (range: 1–182). However, 90.2% of all positive children received praziquantel treatment within two weeks after their initial registration. The median time between the school registration and the household follow-up and water body follow-up was seven days (range: 1–182 days and 0–174 days, respectively). In 89.7% and 86.1%, the household follow-up and water body follow-up, respectively, were conducted within 14 days after the school registration of the children. The median time difference between initial school registration and snail surveys conducted at the water bodies indicated by the positive-tested children was 16 days (range: 0–44 days). In total, 30.0% of the water bodies were surveyed within two weeks after the initial school registration of the children in primary schools or madrassas.

### Passive surveillance in primary health care facilities

In 2021, 19 PHCUs tested 159 patients who reported symptoms of urogenital schistosomiasis for microhematuria as a proxy for *S. haematobium*. Among them, a total of 53 (33.3%) patients were microhematuria-positive. In 2022, 21 PHCUs

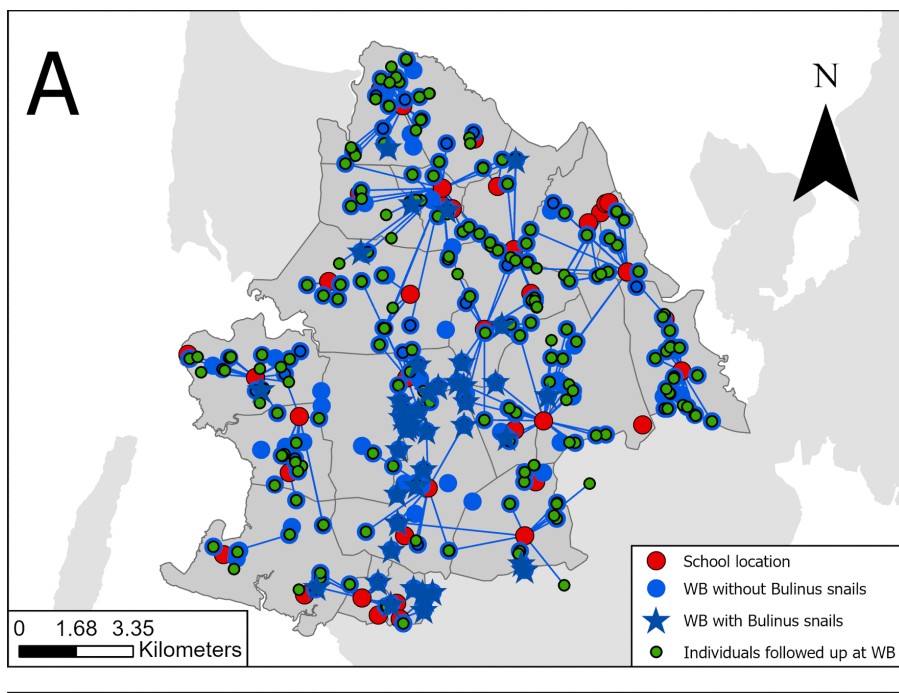

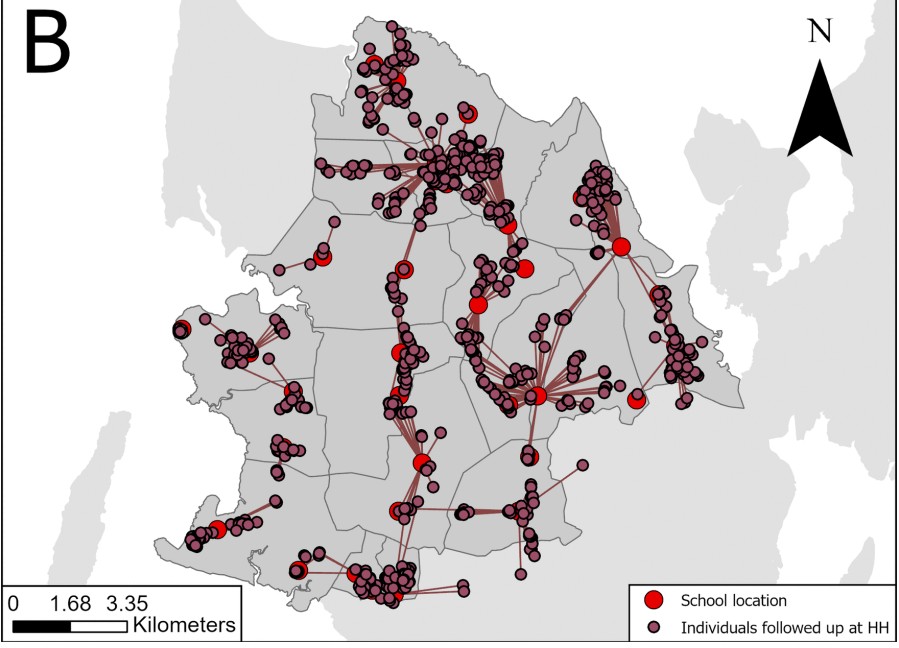

**Fig 2. Locations of active (schools) and reactive (water bodies and households) surveillance of the SchistoBreak study from 2021 to 2023 in the north of Pemba, Tanzania.** The maps show the locations of schools where active surveillance was conducted, and of water bodies (WBs) with and without *Bulinus* (A), and of households (B), where reactive surveillance was conducted. Water bodies and households were identified by tracking positive index cases from schools. The locations of households were geographically masked in order to preserve confidentiality. The image base map (United Republic of Tanzania – Subnational administrative boundaries) was downloaded from the UN Office for the Coordination of Humanitarian Affairs (OCHA) services (https://data.humdata.org/dataset/cod-ab-tza). The data source is: Tanzania National Bureau of Statistics/UN OCHA ROSA. The data are published under the following license: Creative Commons Attribution for Intergovernmental Organizations (CC BY-IGO; (https://creativecommons.org/licenses/by/3.0/igo/legal code)). Additionally, we received written permission to use and adapt the data from OCHA.

**Table 2. Number of water bodies targeted by reactive snail control.**

| Year | Water bodies visited for reactive malacological surveys, N | Water bodies with *Bulinus* presence, N (%) | Water bodies treated with niclosamide, N (%) |
|------|------|------|------|
| **2021** | 112 | 10 (8.9%) | 13 (11.6%) |
| **2022** | 67 | 15 (22.4%) | 19 (28.4%) |
| **2023** | 83 | 10 (12.0%) | 15 (18.1%) |

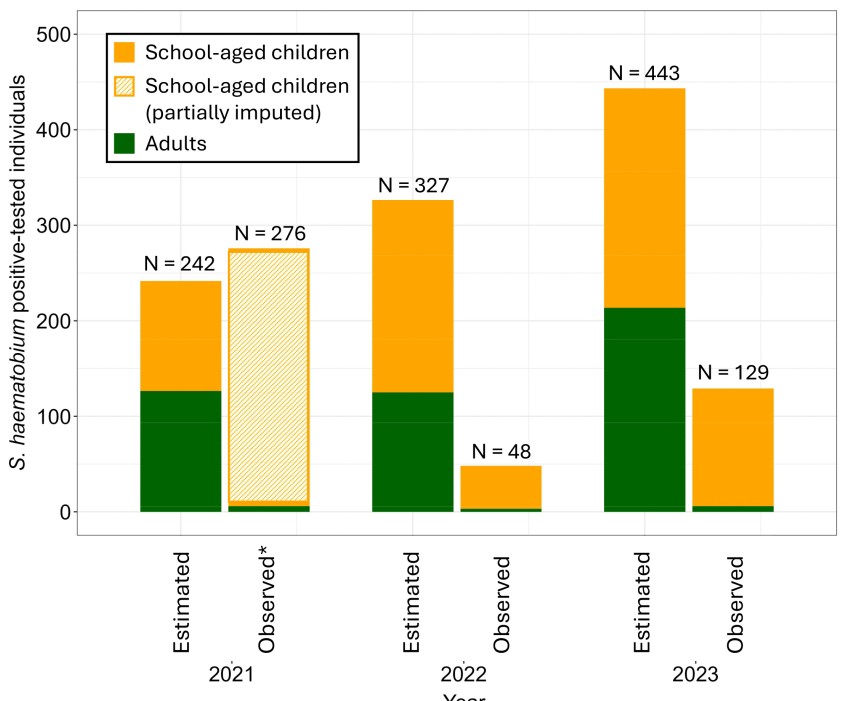

**Fig 3. Estimated number of individuals with *S. haematobium* infection based on annual cross-sectional parasitological surveys and observed number of *S. haematobium* infected individuals during the surveillance intervention period.** * The number of school-aged children identified by surveillance-response activities in 2021 was determined through a combination of diagnostic assessments and data imputation.

**Table 3. Sensitivity of the surveillance interventions per age group, per year, and across all study years.**

| Year | Estimated SAC | Observed SAC | Sensitivity SAC (%) | Estimated adults | Observed adults | Sensitivity adults (%) | Estimated overall | Observed overall | Sensitivity overall (%) |
|------|------|------|------|------|------|------|------|------|------|
| **2021** | 115 | 270* | 234.4 | 126 | 6 | 4.7 | 242 | 276* | 114.2 |
| **2022** | 202 | 45 | 22.3 | 125 | 3 | 2.0 | 327 | 48 | 14.7 |
| **2023** | 230 | 123 | 53.5 | 213 | 6 | 2.8 | 443 | 129 | 29.1 |
| **Overall (2021, 2022 and 2023)** | 547 | 438 | 80.1 | 464 | 15 | 3.2 | 1012 | 453 | 44.8 |
| **Overall (2022 and 2023 only)** | 432 | 178 | 41.2 | 338 | 9 | 2.7 | 770 | 177 | 23.0 |

SAC = school-aged children. * The number of school-aged children identified by the surveillance-response system during active surveillance in schools in 2021 was determined through a combination of diagnostic assessments and data imputation.

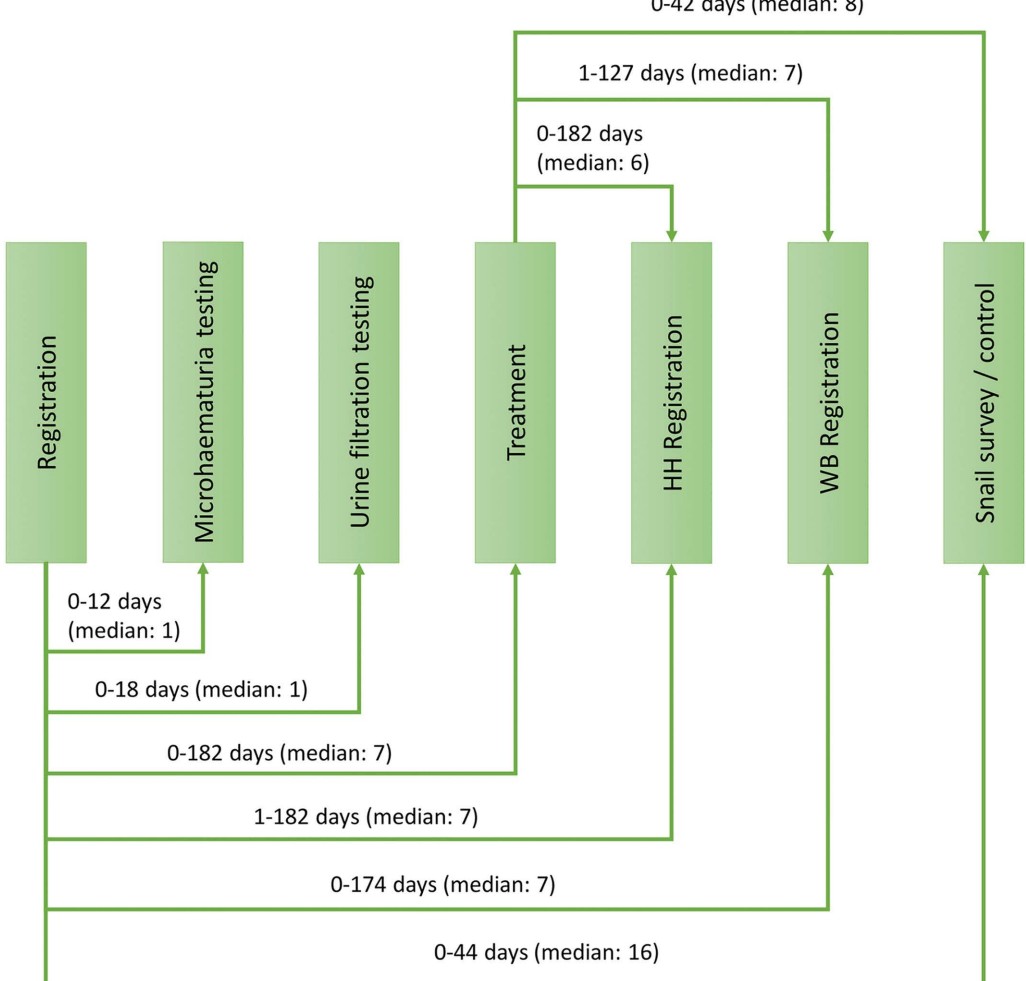

**Fig 4. Time between different steps of the surveillance-response system implemented in low-prevalence implementation units of the SchistoBreak study area from 2021 to 2023.** HH = Household. WB = Water body.

participated in the SchistoBreak study and 130 (16.7%) of 778 individuals with symptoms of urogenital schistosomiasis had microhematuria. In 2023, 23 collaborating PHCUs tested 780 patients for microhematuria, among whom 211 (27.1%) were positive.

### Annual *S. haematobium* prevalence and intensity in low-prevalence implementation units

In the baseline cross-sectional survey conducted in 2021, the *S. haematobium* prevalence in the 11 schools within the 15 low-prevalence areas was 0.5% (7/1552). Among all children, 0.1% (2/1552) had a heavy-intensity infection (Fig 5A). Microhematuria was found in 3.0% (47/1554) of the children (S1 Text and S1A Fig). After the first period of surveillance-response interventions in these areas, the prevalence dropped to 0.4% (6/1653) with 0.0% heavy-intensity infections. In 2022, the *S. haematobium* prevalence in the 14 schools of the 16 low-prevalence areas was 0.6% (12/2123). Among all children, 0.1% (2/2123) had a heavy-intensity infection. After the second period of surveillance-response interventions, the prevalence changed to 0.7% (15/2240) with 0.2% (4/2240) heavy-intensity infections. In 2023, the *S. haematobium*

PLOS Neglected Tropical Diseases

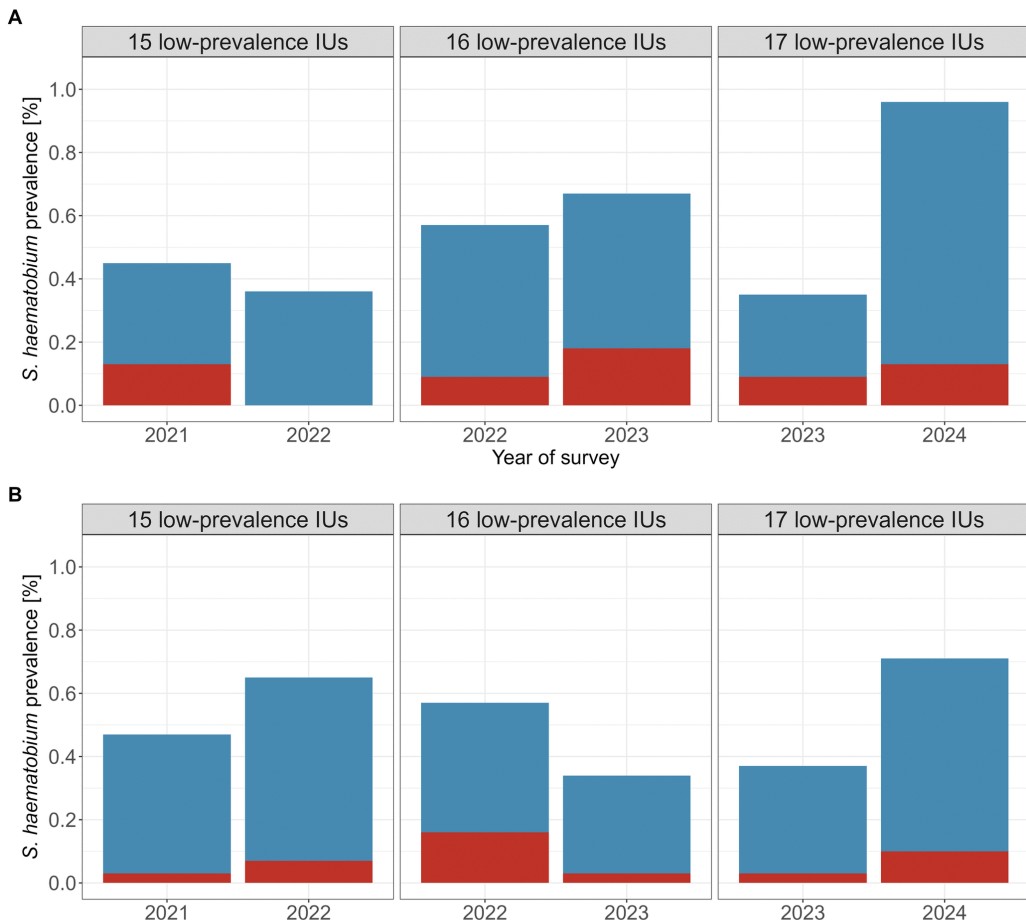

**Fig 5.** *Schistosoma haematobium* **prevalence in low-prevalence implementation units (IUs) in annual school-based surveys (A) and household-based surveys (B) of the SchistoBreak study on Pemba, Tanzania, from 2021 to 2024.** The Fig shows the *S. haematobium* prevalence in low-prevalence IUs in the surveys before and after implementation of a surveillance-response approach. The number of low-prevalence IUs changed annually in line with the SchistoBreak study design. *S. haematobium* assessment was conducted using urine filtration.

prevalence in the 14 schools located in the 17 low-prevalence areas was 0.4% (8/2287), and 0.1% (2/2287) of children had a heavy-intensity infection. After the third surveillance-response intervention period, the prevalence in these areas changed to 1.0% (27/2755) and 0.1% (3/2755) of children had a heavy-intensity infection in 2024.

In the baseline survey conducted in 2021, the *S. haematobium* prevalence in the 15 communities within the 15 low-prevalence areas was 0.5% (14/2969). Among all participants, 0.03% (1/2969) had a heavy-intensity infection (Fig 5B). Microhematuria was detected in 5.5% (162/2969) of the participants (S1 Text and S1B Fig). After the first period of surveillance-response interventions in these areas, the prevalence changed to 0.7% (19/2928) and 0.1% (2/2928) of the tested population had a heavy-intensity infection. In 2022, the *S. haematobium* prevalence in the 16 communities of the 16 low-prevalence areas was 0.6% (18/3175). Among the participants, 0.2% (5/3175) had a heavy-intensity infection. After the second period of surveillance-response interventions, the prevalence dropped to 0.3% (10/2979), and 0.03% (1/2979)

of the participants had heavy-intensity infections. In 2023, the *S. haematobium* prevalence in the 17 communities located in the 17 low-prevalence areas was 0.4% (12/3257), and 0.03% (1/3257) of the participants had a heavy-intensity infection. After the third period of surveillance-response interventions, the prevalence in the 17 communities changed to 0.7% (22/3106) and 0.1% (3/3106) heavy-intensity infections in 2024.

## Discussion

Surveillance, followed up by public health response packages tailored to local transmission settings, is suggested as an effective strategy to achieve the schistosomiasis elimination goals set by the WHO for 2030 [5,16,17]. Our study is among the first studies, if not the first study that implemented and evaluated a surveillance-response approach for schistosomiasis elimination over multiple years, involving several thousand people in Pemba from 2020 to 2024.

Our surveillance-response approach identified a considerable number of *S. haematobium*-infected individuals and water bodies with *Bulinus* that were subsequently treated with praziquantel and niclosamide, respectively. However, the sensitivity, i.e., the proportion of schistosomiasis cases detected by the surveillance system, varied substantially by intervention period. In 2021, likely caused by an incorrect imputation model, the estimated sensitivity was very high (234.4% for school-aged children and hence 114.2% overall for children and adults), and we identified more *S. haematobium* cases than we had estimated from the census data and cross-sectional survey results. In 2022 and 2023, the sensitivity was substantially lower (below 30%). Overall, far fewer *S. haematobium*-infected individuals, in particular adults, were detected by active and reactive surveillance interventions than we should have identified according to the census and survey data. Due to the uncertainty associated with the 2021 imputed data, we consider the overall sensitivity of 23.0% from 2022 and 2023 as accurate. The low number of identified cases may have several reasons. First, we used schoolchildren with positive test results as index cases to identify additional cases in households and at waterbodies. While many additional positive cases were identified through this test-treat-track-test-treat (5T) approach, it may have missed several positive community members who were not related to the index cases [33]. Second, the cases were identified by a single hematuria assessment and/or single urine filtration. Since the sensitivity of these diagnostic approaches is low, particularly in elimination settings, some infections may not have been detected [34,35]. Third, infected individuals may have been missed because our visits in households and at the water bodies did not always coincide with the times people were present at home or used the water bodies. Fourth, a selection bias might have occurred if the schools and madrassas included in the study were not representative of the target population, e.g., in case many non-school-attending children lived in the area. Moreover, a selection bias might have occurred if *S. haematobium*-infected individuals did comply less with study procedures than non-infected participants, e.g., by not providing consent for participation or a urine sample for examination. Finally, individuals with a positive test result in our annual cross-sectional surveys were treated with praziquantel at the end of the survey for ethical reasons, which might have resulted in a lower actual number of cases during the subsequent intervention periods.

Hence, while the surveillance-response approach allowed effective identification and subsequent treatment of positive-tested individuals, its ability to detect all cases in the study area remained very limited. The sensitivity of the approach likely could have been increased with a higher coverage, e.g., by screening more children in more schools in the study area. Our findings are similar to those of malaria research, where a high surveillance coverage was identified as one of the key gaps to be filled on the way to elimination [36,37]. A malaria study conducted in Zanzibar implemented a malaria case notification platform collecting detailed data on all confirmed malaria cases from public and private health facilities, allowing for a prompt testing of household members of positive-tested individuals [38]. For schistosomiasis, no comparative system is in place in Zanzibar, but since urogenital schistosomiasis results in less intense symptoms and morbidity than malaria, and people do not necessarily consult health facilities for treatment, additional active case identification and notification systems are needed to ensure a rapid response and increase the sensitivity of surveillance-response activities.

Indeed, across the four study years, the 23 health facilities involved in the SchistoBreak study for passive surveillance, identified 1738 individuals with symptoms that may be caused by *S. haematobium* infections per study year. Among those, 394 (23.0%) had microhematuria and were hence treated with praziquantel. While these numbers very likely contributed to maintaining the low prevalence and health facilities should be part of an effective schistosomiasis surveillance-response system, they cannot be the only main component for case identification and notification. Of note, for health facilities to play an important role in passive surveillance for schistosomiasis elimination, their capacities need to be improved by regular staff training, and an unbroken supply of (yet unavailable) accurate point-of-care diagnostics and praziquantel for the treatment of cases [39]. As for malaria and similar to our 5T approach, schistosomiasis cases identified in health facilities could also be used as index cases to identify additional cases, as done in a test-and-treat study conducted in Egypt [40].

Rapid response is a critical component of effective surveillance, ensuring timely treatment and reducing potential transmission [41]. The timeliness of our surveillance-response approach was very good. Most cases were identified and treated within one to two weeks. However, while our timeliness was strong, the response was not immediate, as laboratory analyses of samples using urine filtration microscopy required at least one day from sample collection to testing to treatment. A point-of-care test employed directly in schools, households, or at waterbodies would have accelerated the process considerably. Another study conducted in mainland Tanzania, which employed a 5T approach for detecting intestinal schistosomiasis and used the point-of-care circulating cathodic antigen (CCA) test for *S. mansoni* diagnosis, was able to do the testing and treatment on the same day and had an overall timeliness of 52 hours for testing, treating and tracking of 48 index cases and the treatment of 97 follow-up cases [11]. Unfortunately, for *S. haematobium,* no sensitive and specific point-of-care test currently exists [35].

Of note, the implementation of surveillance-response interventions required significant human resources for both active and reactive surveillance, and also for reactive snail control. Good communication and collaboration between the surveillance-response teams were essential for timely follow-up and treatment of cases and waterbodies. Our study area in Pemba consisted of 20 neighboring IUs and covered an area of approximately 150 square kilometers. Moreover, the IUs were connected by a good road system, and schools, villages, water bodies and health facilities were mostly easily accessible by car and reachable within two hours from PHL-IdC. To work on a larger scale, across Pemba or in other countries and areas with different health system capacities, resource levels, or environmental conditions, with good coverage and improved sensitivity, elimination programs will need a large number of well-trained staff and adequate tools for surveillance and response activities, data management, and communication. This need is not unique for schistosomiasis; similar findings have emerged in malaria research, as exemplified in a study from India that demonstrated that recruiting and carefully training health workers from within the study area resulted in independent local capacity for surveillance, vector control, and case management [42,43]. In case surveillance-response is too resource and capacity intense, in countries or communities where large-scale MDA is no longer justified, focal MDA may represent a suitable alternative in areas that have an elevated risk of transmission, as suggested elsewhere [44–46]. Such "hotspot" areas may be identified by regular active or passive surveillance in schools, communities, or health facilities, respectively, or by surveys for and of water bodies containing immediate host snails, since individuals living in areas near water bodies with intermediate host snails have an increased risk of infection [27,39,46–48]. Focal MDA plus insecticiding is also applied as an intervention in malaria control and elimination programs [49,50]. Whether focal MDA without prior testing, but coupled with snail control, may be a resource-oriented and cost-effective intervention for schistosomiasis elimination, remains to be explored. Of note, several studies showed that the effect of snail control can be maximized when applied in peak transmission seasons and that the ideal time for MDA is during the low-transmission season when snail density is close to minimum [51,52]. Hence, by acquiring a more profound understanding of local transmission patterns, the outcomes of future interventions may be further improved.

Our results showed that the implementation of surveillance-response interventions maintained the very low prevalences and infection intensities in our IUs but did not result in interruption of transmission within four years. Besides the low sensitivity of the surveillance-response system to detect and treat cases, also incompliance with praziquantel treatment of infected individuals may have impacted on intervention effectiveness. While infected individuals identified during active and passive surveillance received directly observed treatment at the point-of-care, several infected individuals identified by reactive surveillance were referred to health facilities for treatment. In case these infected individuals did not seek treatment, they may have contributed to maintaining transmission. Such an interruption in the treatment cascade may hence have influenced the outcome in our study and future programmatic implementations should establish robust mechanisms to ensure and verify treatment adherence for all diagnosed cases.

Based on the results of our study, surveillance-response may be considered a useful tool to keep the status quo in the maintenance phase until there is no more risk of a rebound of prevalence and intensity of infection. Whether it is a cost-effective alternative to focal MDA plus snail control in areas with ongoing transmission, or to large-scale MDA at reduced frequency as suggested by WHO remains to be explored. Good coverage and accurate point-of-care diagnostics will be key for improving sensitivity and timelines of surveillance-response, and thus for its successful contribution to elimination efforts. Clearly, to achieve complete interruption of *Schistosoma* transmission, in addition to treatment of infected individuals and mollusciciding against intermediate host snails, measures that prevent reinfection such as improvement of the water, sanitation and hygiene (WASH) infrastructure and sustainable behavior change communication to increase compliance with interventions and treatment will be required [27,53,54].

## Supporting information

**S1 Text. Microhematuria prevalence in low-prevalence implementation units.**
(PDF)

**S1 Fig. Microhematuria prevalence in low-prevalence implementation units.**
(TIF)

**S1 STROBE Checklist. The filled checklist is based on the STROBE Statement-Checklist of items that should be included in reports of observational studies, developed by the STROBE Initiative, https://www.strobe-statement.org/.**
(PDF)

**S1 Dataset. Dataset.**
(CSV)

**S1 Data. Data dictionary.**
(XLSX)

## Acknowledgments

We acknowledge the participants of the SchistoBreak study for providing urine samples, supporting testing and treatment, and sharing invaluable demographic and health-related information. We also thank the children who guided us to the water bodies they frequented. We express our gratitude to the shehas and teachers in our study area for their unwavering support over the years and the many insightful discussions we shared. Additionally, we are indebted to the WHO, Unlimit Health, and the NTD team of the Zanzibar Ministry of Health for their efforts in implementing regular MDAs in Zanzibar and for supplying praziquantel tablets for the surveillance-response approach. Finally, we acknowledge Bayer AG for generously donating niclosamide for our snail control efforts.

## Author contributions

**Conceptualization:** Lydia Trippler, Jan Hattendorf, Sarah Omar Najim, Said Mohammed Ali, Stefanie Knopp.

**Data curation:** Lydia Trippler, Stefanie Knopp.

**Formal analysis:** Lydia Trippler, Jan Hattendorf.

**Funding acquisition:** Stefanie Knopp.

**Investigation:** Lydia Trippler, Mohammed Nassor Ali, Khamis Seif Khamis, Khamis Rashid Suleiman, Said Mohammed Ali, Stefanie Knopp.

**Methodology:** Lydia Trippler, Jan Hattendorf, Stefanie Knopp.

**Project administration:** Lydia Trippler, Shaali Makame Ame, Saleh Juma, Fatma Kabole, Said Mohammed Ali, Stefanie Knopp.

**Resources:** Said Mohammed Ali, Stefanie Knopp.

**Supervision:** Lydia Trippler, Said Mohammed Ali, Stefanie Knopp.

**Visualization:** Lydia Trippler.

**Writing – original draft:** Lydia Trippler, Stefanie Knopp.

**Writing – review & editing:** Lydia Trippler, Jan Hattendorf, Stefanie Knopp.

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
