## [Decision Letter · Decision Letter 0]

23 Sep 2025

Evaluation of surveillance-response interventions for *Schistosoma haematobium* elimination on Pemba Island, Tanzania: a 4-year intervention study with repeated cross-sectional surveys

Dear Dr. Knopp,

Thank you for submitting your manuscript to PLOS Neglected Tropical Diseases. After careful consideration, we feel that it has merit but does not fully meet PLOS Neglected Tropical Diseases's publication criteria as it currently stands. Therefore, we invite you to submit a revised version of the manuscript that addresses the points raised during the review process.

Please submit your revised manuscript within 60 days Nov 22 2025 11:59PM. If you will need more time than this to complete your revisions, please reply to this message or contact the journal office at plosntds@plos.org. Please include the following items when submitting your revised manuscript:

We look forward to receiving your revised manuscript.

Kind regards,

Song Liang

Academic Editor

Francesca Tamarozzi

Section Editor

Shaden Kamhawi

co-Editor-in-Chief

Paul Brindley

co-Editor-in-Chief

**Additional Editor Comments:**

Reviewer #1:

The most pressing issue is the anomalous sensitivity estimate of 114.2% in 2021, which substantially undermines the credibility of the primary outcome and overall study conclusions. The authors must provide a transparent, quantitative explanation for how sensitivity could exceed 100% or revise/remove this calculation. Without resolution, the validity of the results cannot be assured. Closely related is the inconsistent use of diagnostic methods across study years. Although PCR was introduced in 2021, results are presented as aggregated “positive” rates without identifying a clear reference standard or accounting for differences in diagnostic yield. The authors need to define the diagnostic gold standard explicitly, explain how varying methods influenced sensitivity estimates, and ensure that comparisons across years are both transparent and scientifically defensible.

A further concern is the sensitivity calculation’s reliance on an estimated infected population (N_inf) that the authors themselves note may be “overestimated.” This weak foundation makes the sensitivity values questionable. The authors should provide stronger justification for how N_inf was determined, perform sensitivity analyses to show how different assumptions affect the results, and clearly acknowledge limitations if uncertainty remains. Together, these issues represent essential revisions needed to meet PLOS NTD’s standards for methodological rigor, validity, and clarity.

Lastly, the manuscript also briefly mentions different treatment referral pathways (point-of-care versus referral), but their implications for case detection and intervention effectiveness are underdeveloped. Expanding this discussion would help clarify the programmatic impact of these approaches and enhance the manuscript’s contribution to public health practice

Reviewer #2:

The introduction is too brief and should be strengthened by including recent data on the burden of schistosomiasis in sub-Saharan Africa and its contribution to global prevalence, as well as a short history of control in Zanzibar leading up to elimination. The reviewer also requests clarification on why only Pemba was selected for the study, while Unguja—the larger and more populated island—was not included. Relatedly, more explanation is needed on the criteria used to select schools: when the manuscript says the “largest schools” per implementation unit were chosen, does this refer to population size or physical area? Similarly, while the rationale for choosing madrassas is clear, that for schools is not.

Several methodological and programmatic concerns also need to be addressed. The reviewer highlights that patient compliance with praziquantel treatment is often a major barrier to control and asks whether additional initiatives were undertaken to ensure patients actually received treatment. They also raise concerns about the use of chemical molluscicides. Specifically, the authors should explain why niclosamide was applied even in areas where no snails were detected, despite potential ecological risks, and whether any assessment was made of the snail control method’s impact on aquatic ecosystems or steps were taken to prevent human exposure when communities use these waters for domestic purposes. The manuscript also mentions that environmental data (water chemistry, vegetation type) were collected, but these variables are not discussed in relation to snail presence or abundance and should be incorporated.

Finally, the reviewer requests clarification on how the outcome of passive surveillance interventions was measured (e.g., whether diagnosed cases increased following the intervention). They also note missing prevalence data for primary school children in 2022 (line 389), which should be included. Addressing these points will strengthen the manuscript’s methodological rigor and provide important context for interpreting the intervention results

**Journal Requirements:**

At this stage, the following Authors/Authors require contributions: Said Mohammed Ali. Please ensure that the full contributions of each author are acknowledged in the "Add/Edit/Remove Authors" section of our submission form.

3) Some material included in your submission may be copyrighted. According to PLOSu2019s copyright policy, authors who use figures or other material (e.g., graphics, clipart, maps) from another author or copyright holder must demonstrate or obtain permission to publish this material under the Creative Commons Attribution 4.0 International (CC BY 4.0) License used by PLOS journals. Please closely review the details of PLOSu2019s copyright requirements here: PLOS Licenses and Copyright. If you need to request permissions from a copyright holder, you may use PLOS's Copyright Content Permission form.

Potential Copyright Issues:

i) Figure 2. Please (a) provide a direct link to the base layer of the map (i.e., the country or region border shape) and ensure this is also included in the figure legend; and (b) provide a link to the terms of use / license information for the base layer image or shapefile. We cannot publish proprietary or copyrighted maps (e.g. Google Maps, Mapquest) and the terms of use for your map base layer must be compatible with our CC BY 4.0 license.

4) We note that your Data Availability Statement is currently as follows: "All relevant data are within the manuscript and its Supporting Information files.". Please confirm at this time whether or not your submission contains all raw data required to replicate the results of your study. Authors must share the “minimal data set” for their submission. PLOS defines the minimal data set to consist of the data required to replicate all study findings reported in the article, as well as related metadata and methods (https://journals.plos.org/plosone/s/data-availability#loc-minimal-data-set-definition).

**Reviewers' Comments:**

Reviewer's Responses to Questions

**Key Review Criteria Required for Acceptance?**

**Methods**

-Are the objectives of the study clearly articulated with a clear testable hypothesis stated?

-Is the study design appropriate to address the stated objectives?

-Is the population clearly described and appropriate for the hypothesis being tested?

-Is the sample size sufficient to ensure adequate power to address the hypothesis being tested?

-Were correct statistical analysis used to support conclusions?

-Are there concerns about ethical or regulatory requirements being met?

Reviewer #1: (No Response)

Reviewer #2: (No Response)

**Results**

-Does the analysis presented match the analysis plan?

-Are the results clearly and completely presented?

-Are the figures (Tables, Images) of sufficient quality for clarity?

Reviewer #1: (No Response)

Reviewer #2: (No Response)

**Conclusions**

-Are the conclusions supported by the data presented?

-Are the limitations of analysis clearly described?

-Do the authors discuss how these data can be helpful to advance our understanding of the topic under study?

-Is public health relevance addressed?

Reviewer #1: (No Response)

Reviewer #2: (No Response)

**Editorial and Data Presentation Modifications?**

Reviewer #1: (No Response)

Reviewer #2: (No Response)

**Summary and General Comments**

Reviewer #1: (No Response)

Reviewer #2: (No Response)

PLOS authors have the option to publish the peer review history of their article (what does this mean? ). If published, this will include your full peer review and any attached files.

**Do you want your identity to be public for this peer review?** For information about this choice, including consent withdrawal, please see our Privacy Policy .

Reviewer #1: No

Reviewer #2: **Yes:** Nicolaus Omary Mbugi

**Figure resubmission:**
---

## [Decision Letter · Decision Letter 1]

17 Dec 2025

*Schistosoma haematobium*
Response to Reviewers
Revised Manuscript with Track Changes
Manuscript

Shaden Kamhawi

co-Editor-in-Chief

Paul Brindley

co-Editor-in-Chief

**Journal Requirements:**

**Reviewers' comments:**

**Key Review Criteria Required for Acceptance?**

**Methods**

-Are the objectives of the study clearly articulated with a clear testable hypothesis stated?

-Is the study design appropriate to address the stated objectives?

-Is the population clearly described and appropriate for the hypothesis being tested?

-Is the sample size sufficient to ensure adequate power to address the hypothesis being tested?

-Were correct statistical analysis used to support conclusions?

-Are there concerns about ethical or regulatory requirements being met?

Reviewer #1: (No Response)

Reviewer #2: (No Response)

**Results**

-Does the analysis presented match the analysis plan?

-Are the results clearly and completely presented?

-Are the figures (Tables, Images) of sufficient quality for clarity?

Reviewer #1: (No Response)

Reviewer #2: (No Response)

**Conclusions**

-Are the conclusions supported by the data presented?

-Are the limitations of analysis clearly described?

-Do the authors discuss how these data can be helpful to advance our understanding of the topic under study?

-Is public health relevance addressed?

Reviewer #1: (No Response)

Reviewer #2: (No Response)

**Editorial and Data Presentation Modifications?**

Reviewer #1: (No Response)

Reviewer #2: (No Response)

**Summary and General Comments**

Reviewer #1: 1.The 2021 sensitivity estimate of 114.2% is anomalously high due to data imputation, which undermines the credibility of the three-year overall performance assessment (44.8%). To address this, the manuscript should clearly present the 2022-2023 sensitivity (23.0%) – derived from complete data – as the primary finding in the Results section. Furthermore, the Discussion should explicitly state the uncertainty associated with the 2021 imputed data and critically examine the potential biases introduced by estimating the total number of infections (N_inf) from projected census and survey prevalence data in a low-transmission setting.

2, The study's conclusions are drawn from a specific context in Pemba Island, and their applicability to other low-prevalence regions requires more thorough discussion. To strengthen the paper, a dedicated subsection should be added to the Discussion that outlines the contextual factors of the study setting and thoughtfully discusses the potential adjustments needed for implementing this surveillance-response approach in areas with different health system capacities, resource levels, or environmental conditions.

3.A significant limitation is the lack of data on final treatment completion for laboratory-confirmed cases who were referred to health facilities, which weakens the analysis of why transmission interruption was not achieved. This gap should be explicitly acknowledged in the Limitations section of the Discussion. The authors should discuss how this potential break in the treatment cascade could have influenced the outcome and recommend that future programmatic implementations establish mechanisms to verify treatment adherence for all diagnosed cases.

Reviewer #2: (No Response)

PLOS authors have the option to publish the peer review history of their article (what does this mean? ). If published, this will include your full peer review and any attached files.

**Do you want your identity to be public for this peer review?** For information about this choice, including consent withdrawal, please see our Privacy Policy .

Reviewer #1: No

Reviewer #2: **Yes:** Nicolaus Omari Mbugi

**Figure resubmission:**

**Reproducibility:** To enhance the reproducibility of your results, we recommend that authors of applicable studies deposit laboratory protocols in protocols.io, where a protocol can be assigned its own identifier (DOI) such that it can be cited independently in the future. Additionally, PLOS ONE offers an option to publish peer-reviewed clinical study protocols. Read more information on sharing protocols at https://plos.org/protocols?utm_medium=editorial-email&utm_source=authorletters&utm_campaign=protocols

---

## [Editor Report · Decision Letter 2]

19 Jan 2026

Dear Dr Knopp,

We are pleased to inform you that your manuscript 'Evaluation of surveillance-response interventions for *Schistosoma haematobium* elimination on Pemba Island, Tanzania: a 4-year intervention study with repeated cross-sectional surveys' has been provisionally accepted for publication in PLOS Neglected Tropical Diseases.

Best regards,

Francesca Tamarozzi

Section Editor

Francesca Tamarozzi

Section Editor

Shaden Kamhawi

co-Editor-in-Chief

Paul Brindley

co-Editor-in-Chief

---

## [Editor Report · Acceptance letter]

Dear Dr Knopp,

We are delighted to inform you that your manuscript, "Evaluation of surveillance-response interventions for *Schistosoma haematobium* elimination on Pemba Island, Tanzania: a 4-year intervention study with repeated cross-sectional surveys," has been formally accepted for publication in PLOS Neglected Tropical Diseases.

Best regards,

Shaden Kamhawi

co-Editor-in-Chief

Paul Brindley

co-Editor-in-Chief
